# Rank-Rankl-Opg Axis in Multiple Sclerosis: The Contribution of Placenta

**DOI:** 10.3390/cells11081357

**Published:** 2022-04-15

**Authors:** Sofia Passaponti, Leonardo Ermini, Giulia Acconci, Filiberto Maria Severi, Roberta Romagnoli, Santina Cutrupi, Marinella Clerico, Gisella Guerrera, Francesca Ietta

**Affiliations:** 1Department of Life Sciences, University of Siena, 53100 Siena, Italy; sofia.passaponti@student.unisi.it (S.P.); leonardo.ermini@unisi.it (L.E.); roberta.romagnoli@unisi.it (R.R.); 2Department of Molecular and Developmental Medicine, Division of Prenatal Diagnosis and Obstetrics, University of Siena, 53100 Siena, Italy; giulia.acconci@student.unisi.it (G.A.); filiberto.severi@unisi.it (F.M.S.); 3Department of Clinical and Biological Sciences, University of Turin, 10124 Turin, Italy; santina.cutrupi@unito.it (S.C.); marinella.clerico@unito.it (M.C.); 4Neuroimmunology Unit, IRCCS Fondazione Santa Lucia, 00179 Rome, Italy; g.guerrera@hsantalucia.it

**Keywords:** pregnancy, placenta, osteoprotegerin, autoimmune diseases

## Abstract

Women with multiple sclerosis (MS) can safely become pregnant and give birth, with no side effects or impediments. Pregnancy is generally accepted as a period of well-being in which relapses have a softer evolution, particularly in the third trimester. Herein, we hypothesized that the placenta, via its “secretome”, could contribute to the recognized beneficial effects of pregnancy on MS activity. We focused on a well-known receptor/ligand/decoy receptor system, such as the one composed by the receptor activator of nuclear factor-kB (RANK), its ligand (RANKL), and the decoy receptor osteoprotegerin (OPG), which have never been investigated in an integrated way in MS, pregnancy, and placenta. We reported that pregnancy at the term of gestation influences the balance between circulating RANKL and its endogenous inhibitor OPG in MS women. We demonstrated that the placenta at term is an invaluable source of homodimeric OPG. By functional studies on astrocytes, we showed that placental OPG suppresses the mRNA expression of the CCL20, a chemokine responsible for Th17 cell recruitment. We propose placental OPG as a crucial molecule for the recognized beneficial effect of late pregnancy on MS and its potential utility for the development of new and more effective therapeutic approaches.

## 1. Introduction

Epidemiologic and clinical evidence suggests a beneficial effect of pregnancy in women with multiple sclerosis (MS), a long-lasting and immune-mediated disorder affecting the central nervous system (CNS) [1,2].

During pregnancy, the disease has a softer evolution, the number and intensity of the relapses decrease over the nine months of gestation, particularly in the last trimester. However, this temporary beneficial effect is followed by a flaring up of the disease in the early post-partum period, resuming its course soon after delivery [1,3].

It is also recognized that MS does not affect pregnancy outcomes; indeed, compared with healthy women, maternal MS does not put patients at higher risk for miscarriage, preeclampsia, and premature birth [4,5]. Furthermore, obstetric outcomes such as the type of delivery, gestational age, and birth weight in MS pregnancy are similar to those of the general population [6,7].

Substantial experimental evidence suggests that the protective effect of pregnancy may be related to the immunological changes that occur during normal pregnancy, allowing tolerance to fetal antigens and to autoimmune targets though preventing pathogen infections [8].

While most of the studies focused their attention on the contribution of the maternal immune response, very few have looked at the role of the placenta, the organ interposed between the mother and the fetus with a fundamental role for pregnancy, maternal and fetal health [9,10,11].

The placenta is an important source of a wide range of biologically active factors including steroids, cytokines, pregnancy-associated glycoproteins, cell-free nucleic acids, and extracellular vesicles [12,13,14,15]. The unexpectedly rich pool of placental secreted factors also includes the production of decoy receptors able to recognize inflammatory molecules with high affinity and specificity, thus fine-tuning maternal inflammatory responses [16]. In a mouse model of autoimmune hypophysitis, a disease known to be strongly associated with pregnancy, the placenta produces soluble Tumor necrosis factor receptor 1 (sTNF-R1), a counter-regulatory molecule that attenuates the proinflammatory responses induced by TNF-α [11].

In the current study, we aimed to examine the involvement of the placenta in the production and release of the soluble decoy receptor osteoprotegerin (OPG), a member of the TNF receptor superfamily [17], able to control the binding between receptor activator of nuclear factor-kB (RANK) and its ligand (RANKL), never investigated before in the context of MS, pregnancy, and placenta.

RANK is a type I transmembrane protein, and the ligation of this receptor by RANKL leads to cellular downstream signaling that promotes translocation and activation of transcription factors including Nuclear Factor Of Activated T Cells 1 (NFATc1), cAMP response element-binding protein (CREB), Nuclear Factor Kappa B (NFkB), Activator protein 1 (AP-1), and Microphthalmia-associated transcription factor (MITF) depending on the cell type [18]. Signaling via RANK can be induced by either membrane-bound RANKL or its soluble form (sRANKL) derived from proteolytic cleavage of the membrane-bound protein or alternative mRNA splicing. The endogenous inhibitor of RANKL–RANK interactions is the soluble decoy receptor OPG that targets both the soluble and the membrane-bound forms of RANKL [19].

Studies on this system have highlighted its profound involvement in macrophage polarization and T-cell biology including differentiation, activation, and recruitment [19,20]. In addition, a dysregulation of the RANK-RANKL-OPG system has been associated with loss of T cell tolerance and risk of autoimmune disease [21]. The triad is also implicated in several aspects of neural tissue damage and reparative processes driven by interactions between immune cells and several types of cells within the CNS [22].

Herein, we reported that pregnancy at term of gestation influences the balance between sRANKL and its endogenous inhibitor OPG in MS women. We also demonstrated that the placenta at term produces high level OPG in its homodimeric form, known to exert the highest activity as a decoy receptor.

Moreover, functional studies on astrocytes showed that placental OPG suppresses cell response in the secretion of CLL20, a chemokine responsible for Th17 cell recruitment.

Therefore, our study suggests that placental OPG may be a crucial molecule for the modulation of the crosstalk between CNS and the immune system of the mother, and for the recognized beneficial effect of late pregnancy in MS patients.

## 2. Materials and Methods

### 2.1. Placental Tissues from Healthy Pregnant Women

Placental tissues were obtained at the Prenatal Diagnosis and Obstetrics Division of the Department of Molecular and Developmental Medicine (University Hospital of Siena, Siena, Italy). Samples at first-trimester pregnancy (*n* = 10), were collected from normal undergoing pregnancy terminated for psychological reasons by dilatation and curettage. Samples at term of gestation (*n* = 10) were collected from elective caesarean section planned for cephalic presentation of the fetus or previous caesarean. Clinically diagnosed infectious diseases, pregnancy disorders such as pre-eclampsia, HELLP (Hemolysis, Elevated Liver enzymes and Low Platelets) syndrome, gestational diabetes and genetic deficiencies of the fetus were exclusion criteria. All samples immediately after collection were in part snap-frozen and stored at −80 °C, in part, formalin fixed and in part processed for chorionic placenta explant cultures within one hour after collection.

### 2.2. Serum Collection and Processing

Blood samples were collected during routine check-up and centrifuged within 30 min from sampling (6000× *g* for 10 min, at 4 °C), the supernatants were collected and stored at −20 °C until assayed.

The donors, referred to the Obstetrics and Gynecology Division of the Department of Molecular and Developmental Medicine (University Hospital of Siena, Siena, Italy), included healthy women with physiological pregnancy during the first-trimester of pregnancy (*n* = 10; 1st Trimester), at term of gestation (*n* = 20; Term) and sex and age matched non-pregnant healthy donors (*n* = 10; NP) enrolled as the control group. Umbilical cord blood samples were also collected at delivery from newborns (*n* = 10; Cord serum) at the time of caesarean section.

Pre-existing serum samples, collected at the academic neurological unit, Department of Clinical and Biological Sciences, University of Turin, Italy included MS patients at term of gestation (Term) and one month after delivery (post-partum: PP) (*n* = 10) with clinically defined RRMS.

Pre-existing serum samples collected at Neuroimmunology Unit, IRCCS Fondazione Santa Lucia, Rome, Italy, included MS women non-pregnant (*n* = 19; NP) of reproductive age with clinically defined RRMS. Clinical data of the study population, healthy women Table 1 and MS women Table 2. 

### 2.3. Human Placenta Explant Culture and Treatment

After several washes with sterile phosphate-buffered saline (PBS), chorionic villous explants from placental tissues (*n* = 4, placentae from first trimester of gestation; *n* = 10 placentae from term of gestation) were established as described by Caniggia with minor modifications [23]. Explants were incubated overnight at 37 °C in an atmosphere of 3% O_2_, 5% CO_2_ for first-trimester explants or 8% O_2_, 5% CO_2_ for term explants in DMEM F12 (phenol red and serum-free) plus L-glutamine (Gibco, Invitrogen, Basel, Switzerland and 1% antibiotics (penicillin-streptomycin). Medium was then replaced with a fresh one and placental explants cultured for 48 h.

*n* = 3 placentae from term of gestation were also used to establish placental explants for hormone exposure experiments. After the overnight incubation, term placental explants were treated with hormones according to their physiological levels found in the maternal serum at the end of gestation [24]: 1 × 10^−8^ M Estradiol (E2), 7 × 10^−8^ M Estriol (E3) and 1 × 10^−6^ M Progesterone (Pg) (Sigma Chemical Co. St. Louis, MO, USA) for 48 h.

At the end of experiments, explant cultures and their respective supernatants were collected and processed for protein analyses or subsequent analysis.

### 2.4. Astrocytes Cell Culture and Treatment

The human astrocytic U-373 MG cell line was grown in DMEM supplemented with 10% FBS, 1% L-glutamine, and 1% penicillin/streptomycin antibiotics. The cells were maintained at 37 °C in a humidified 5% CO_2_ atmosphere.

U373 MG cells were cultured in six-well plates (2.5 × 10^5^ cells/mL) in complete medium for 24 h at the same condition. Next, cells were treated or not (controls) with 100 ng/mL of RANKL alone or (R&D System Inc., Minneapolis, MN, USA) in combination with 25, 50 or 100 ng/mL of OPG (R&D System Inc.) according to Shang et al. [25]. 

After additional 24 h of incubation, cells were processed for mRNA expression analysis. Treatments and controls were carried out in quadruplicate.

### 2.5. Immunoprecipitation

Term explants supernatants (from *n* = 4 placentae) were employed in immunoprecipitation experiments. Then, 10 mL of supernatants for each placenta were harvested, and 5 mL of each sample were incubated with anti-human OPG antibodies conjugated with Sepharose beads (SantaCruz Biotechonology, Dallas, TX, USA) overnight at 4 °C, with constant rocking. Supernatants depleted from OPG were analyzed by ELISA as described below.

### 2.6. RNA Isolation and qRT-PCR

Total RNA from first-trimester (*n* = 10 placentae) and term placental tissues (*n* = 10 placentae) or U373 cell culture (*n* = 3) was isolated using the Zymo-Spin I Fast-Spin column (Zymo-Research, Irvine, CA, USA) according to manufacturer’s instructions. Total RNA was quantified using the Nanodrop 1000 Spectrophotometer (Thermo Fisher Scientific, Waltham, MA, USA). RNA was reversed transcribed into cDNA using the High-Capacity cDNA Reverse Transcription Kit (Applied Biosystems Group, Foster City, CA, USA). RT-PCR reactions were run in the StepOne™ Real-Time PCR System instrument (Applied Biosystems Group). cDNA was amplified using the iQTM SYBR^®^ Green Supermix (Bio-Rad, Cambridge, MA, USA). Pre-developed primers for OPG, IL-1α, IL-6, TNF-α, CCL20 and CCL2 (Bio-Rad) were used for relative quantification of mRNA; GAPDH and 18 S (Bio-Rad) were used as housekeeping. The expression level of the selected genes was calculated by the 2^−ΔΔCt^.

### 2.7. Western Blot Analysis

Total proteins from placental samples (*n* = 10 for first trimester and *n* = 10 for term placentae) or chorionic villous explants were obtained by mechanical homogenization with nitrogen. SDS-PAGE and non-reducing Western blot analysis were performed using 40 μg of total protein lysates and explants supernatants, respectively. Primary antibodies were mouse monoclonal anti-human OPG (1:1000, Listarfish, Milano, Italia), anti-human RANK (1:1000; R&D System Inc., Minnneapolis, MN, USA) and mouse anti-human β-actin (1:2000, Santa Cruz Biotechnology, Dallas, TX, USA) antibodies. Appropriate horseradish peroxidase-conjugated secondary antibody (1:6000) was used (Bio-Rad) and chemiluminescence signals captured using by ChemiDoc (Bio-Rad).

### 2.8. Immunohistochemistry

Immunohistochemistry was performed on 4 μm thick sections of formalin-fixed, paraffin-embedded first-trimester and term placental tissues (*n* = 3 for each period of gestation). Mouse monoclonal antibody against OPG (Listarfish) was used at 1:2000 dilution and secondary antibody was used according to the manufacturer’s instructions (Sigma Chemical Co., St. Louis, MO, USA). Slides were counterstained using Mayer’s hematoxylin. Negative controls were carried out by substituting the primary antibody by the appropriate normal isotype antibody. Negative controls are shown as Appendix A.

### 2.9. Immunofluorescence

For immunofluorescence analysis, U373 cells (*n* = 3 separate experiments) were grown on coverslips, washed with PBS, and fixed for 10 min in 4% paraformaldehyde and then permeabilized or not in 0.1% Triton X-100 in PBS for 5 min. The slides were incubated for 1 h at room temperature with the primary mouse anti-human RANK antibody (1:100; R&D System Inc.). The bound antibody was revealed by incubation with Alexa Fluor 488-labelled goat anti-mouse IgG antibody (Thermo Fisher Scientific). The slides were mounted with Fluoromount Aqueous Mounting (Sigma Chemical Co.) to avoid fading of fluorescence. Images were acquired on Zeiss LSM700 confocal microscope.

### 2.10. ELISA

ELISA for OPG and sRANKL was performed in Serum samples and explants supernatants using Human DuoSet ELISA kits (R&D systems Inc.), according to the manufacturer’s instructions. The sensitivity limit of the OPG ELISA assay was 43 pg/mL with intra- and inter-assay coefficients of variation of 2.18% ± 0.95% and 4.71% ± 0.47%, respectively. The sensitivity limit of the sRANKL ELISA assay was 85 pg/mL with intra- and inter-assay coefficients of variation of 3.07% ± 0.86% and 6.12% ± 0.73, respectively. ELISA assay was performed in duplicate for each sample analyzed.

### 2.11. Statistical Analyses

All data sets from the experiments were analyzed for normality using Shapiro–Wilk or Kolmogorov–Smirnov followed by either parametric or non-parametric tests as appropriate. Statistical analyses were performed using GraphPad Prism 8 (GraphPad Software, San Diego, CA, USA). Densitometric analyses were completed using ImageJ software. The level of significance was set at *p* < 0.05.

## 3. Results

### 3.1. OPG and sRANKL in Serum of MS-Affected and Healthy Pregnant and Non-Pregnant Women

To determine the physiological relevance of pregnancy in the functionality of the RANK-RANKL-OPG triad in MS, we first evaluated the circulating levels of the OPG and sRANKL in serum samples collected from pregnant and non-pregnant MS and healthy women (Figure 1).

#### 3.1.1. Pregnancy Increases OPG Serum Levels and Differentially Regulates Srankl in MS-Affected and Healthy Women

Serum samples from MS patients were tested at term in pregnancy, the time in which relapses decrease significantly, one month after delivery when a recrudescence of the disease takes place and in reproductive age not in pregnancy (Figure 1A,B). Samples in healthy women were tested in pregnancy at the first-trimester, at term of pregnancy, and in reproductive age not in pregnancy (Figure 1C,D).

Results in MS patients showed that OPG serum levels were significantly higher in term pregnancy than in the post-partum (*p* = 0.0024) or the non-pregnant period (*p* = 0.0025) (Figure 1A). In contrast, no change in sRANKL serum level was detected among the three study groups (Figure 1B).

According to published data [26], quantification analysis in healthy women showed significantly higher levels of OPG in serum at term of pregnancy in comparison to the first-trimester or the non-pregnant period (Figure 1C). Serum sRANKL levels were significantly higher level during pregnancy in comparison to the non-pregnant period, with no difference between the two gestational periods (Figure 1D).

#### 3.1.2. Pregnancy Influences the Balance between Srankl and Opg in Ms-Affected Women, Returning It to the Conditions of Healthy Subjects

Since pregnancy influences serum OPG and sRANKL and the disturbance of the sRANKL to OPG ratio rather than alterations in their absolute levels, has been associated with many disorders including autoimmune diseases [22,27] we considered the sRANKL/OPG ratio in pregnant and non-pregnant women with or without MS (Figure 1E) as a marker of sRANKL bioavailability.

The results showed that MS disease is characterized by a high sRANKL and OPG ratio in non-pregnant women, which decreased in pregnancy at term, reaching levels of healthy pregnant women (Figure 1E). These changes reflected the robust increase of circulating OPG in MS women at term of pregnancy and probably a lower availability of sRANKL.

#### 3.1.3. OPG in Maternal and Cord Blood Serum

We then quantified the OPG protein in paired serum from maternal and cord blood at term of pregnancy and, we found a significantly higher level in the maternal than in cord serum samples (Figure 1F).

### 3.2. Human Placenta Is an Invaluable Source of OPG

To establish whether the placenta is a source of OPG during pregnancy we performed a study in healthy women, at first-trimester and term pregnancy (Figure 2).

#### 3.2.1. OPG Production and Distribution in Placental Tissues

mRNA expression analysis by qRT-PCR for OPG in placental samples revealed significantly higher levels of OPG in term placenta compared to tissues collected within the first-trimester of pregnancy (Figure 2A). However, OPG protein analysis showed no differences in intra-tissue content between the two gestational periods. Under reducing conditions, a single band of approximately 60 kDa, corresponding to the molecular weight of the monomeric form of the protein, was evident in all samples examined with no significant difference as supported by the densitometric analysis (Figure 2B upper and lower panels).

To assess OPG spatial distribution in the human placenta, tissues were examined by immunohistochemistry (Figure 2C).

The first-trimester placenta showed strong immunoreactivity for OPG in the villous stromal cells and the perivascular cells of fetal capillaries within the villi core. OPG immunoreactivity was also observed in the cytotrophoblasts, the innermost trophoblast cellular layer of the placenta, while the outermost layer the syncytiotrophoblast, in direct contact with the maternal blood, showed no immunoreactivity (Figure 2C, upper panels).

In term placenta, OPG was localized in the villous cytotrophoblast cells, which are at this gestational age in closer proximity to the maternal blood with respect to first-trimester ones; OPG immunoreactivity in the villous stroma and in the perivascular cells of fetal capillaries was drastically decreased compared to first-trimester tissues (Figure 2C, lower panels).

#### 3.2.2. OPG Is Highly Released by Placenta Explant Cultures Established from Tissues at Term of Pregnancy

To further define the contribution of the first-trimester and term placenta to the release of OPG under basal conditions, we performed in vitro cultures of human villous explant tissues from healthy women, collected at the two gestational ages. The placental cultures were maintained under physiological oxygen tension corresponding to 3% of oxygen for cultures established from first-trimester tissues and 8% of oxygen for cultures established from tissues at term of gestation [28] (Figure 3).

Results from in vitro experiments showed that after 48 h of culture, the conditioned media from term placenta explants was significantly enriched by OPG (Figure 3A).

OPG is synthesized as 55–60 kDa monomer within the cell, eventually converted to a disulfide-linked homodimer, and finally released outside the cells. Monomeric and homodimeric OPG are both released and active, however, the disulfide-linked dimer has a higher binding affinity for RANKL exerting more potent biological activity both in in vivo and in vitro [29,30]. Since none of the commercially available ELISA tests discriminate between the monomeric and homodimeric form of the OPG, we processed the placenta explants conditioned media for protein analysis under non-reducing conditions. Results showed the presence of an intense band at 120 kDa (corresponding to the molecular weight of the OPG homodimer) and a slight one at 60 kDa (corresponding to the monomeric form of the protein) in term samples (Figure 3B). In contrast, the first-trimester samples showed the presence of a 60 kDa band only after an overexposure of the membranes (data not shown) indicating the release of the protein, albeit at low levels, in accordance with the ELISA data but only in the monomeric form.

To establish factors regulating placenta OPG production, we next performed in vitro experiments exposing the cultures established from term placenta tissues to the main hormones characterizing the pregnant environment at term of gestation. We investigated the effect of oestrogens including E2, E3, and that of the steroid hormone Pg at concentrations corresponding to those found physiologically in the maternal circulation at the end of pregnancy (E2 at 1 × 10^−8^ M; E3 at 7 × 10^−8^ M; Pg at 1 × 10^−6^ M) [24]. All treatments increased OPG production by placenta cultures; however, only the exposure with estrogens was found to be significant relative to untreated control cultures (Ct vs. E2, *p* < 0.01 and Ct vs. E3, *p* < 0.001). The effect of E3 treatment on OPG release resulted also significant compared to E2 (Figure 3C).

The uniqueness of the term placenta in producing the homodimeric OPG was further evaluated in lysates from several cells by Western blot under non-reducing conditions (Appendix A).

The results pointed out that the term placenta predominantly expressed the homodimeric form at markedly higher levels than other cells, thus revealing its extraordinary capacity to produce the OPG in the most active molecular structure.

### 3.3. Role of RANK-RANKL-OPG System in Astrocytes

Our findings on sRANKL/OPG ratio in pregnant MS patients and placental OPG production, prompted us to investigate whether the OPG, and in particular, the OPG produced by the placenta could exert its biological activity on maternal cells relevant to the improvement of MS during pregnancy, i.e., astrocytes that could be challenged in the course of MS by the high levels of circulating sRANKL.

#### 3.3.1. Astrocytes Activation by RANKL

We first examined the potential of astrocytes in RANK-RANKL-OPG system activity (Figure 4). Immunofluorescence performed in permeabilized or non-permeabilized astrocytoma cells showed the presence of RANK protein in the cytoplasm and at the membrane level, suggesting their suitability to test placental OPG activity on RANK-RANKL system (Figure 4A). At the same time, no effect on the expression level of RANK was observed after the treatment of astrocytes cultures with recombinant human RANKL (rh RANKL) (R&D Systems) (100 ng/mL) (Figure 4B).

To investigate the effect of high sRANKL levels exposure on astrocytes as it may occur in vivo, in subsequent experiments, the cells were subjected for 24 h to rh RANKL (R&D Systems) (100 ng/mL) and the total mRNA used to evaluate by qRT-PCR the expression levels of the pro-inflammatory cytokines IL-1β, IL-6, TNF-α and two C-C type chemokine ligands, i.e., CCL20 and CCL2 well known factors for T cells recruitment [31].

Results showed that the RANK-RANKL system activation produced a significant up-regulation in the mRNA expression of the CCL20, a similar trend was observed for CCL2 and IL-1β although not significant. As concerns IL-6, the treatment did not result in any changes, while a decrease, though non-significant, was observed for TNF-α (Figure 4C). 

#### 3.3.2. The Role of Placental OPG in Astrocyte Activation

These results inspired us to examine the effect of conditioned media (CM) from placental explants at term of gestation that we had proven to be highly enriched in homodimeric OPG on astrocytes. The CM from every single experiment was in part processed to obtain a CM OPG-depleted via immunoprecipitation.

The whole placenta-derived CM and its OPG-depleted counterpart were assayed for OPG and sRANKL by ELISA. Quantification analysis revealed a reduction greater than 40% (median 44.69; minimum 34.36; 25% Percentile 36.05; 75% Percentile 55.26; Maximum 57.60) for OPG after depletion (Appendix A) and undetectable values for sRANKL, which resulted under the limit of detection in all CM tested.

The astrocytes exposed to rh RANKL plus placenta-derived CM or placenta-derived CM OPG-depleted were finally tested for CCL20 mRNA expression levels.

We found that OPG depletion produced a significant and robust up regulation (threefold rise) of CCL20 mRNA as compared to astrocytes exposed to rh RANKL plus the whole placenta-derived CM or only CM (Figure 4D).

Finally, to confirm the activity of placenta OPG on RANK activation and CCL20 expression, we treated the cells with rh RANKL in combination with rh OPG (R&D Systems) (25, 50 and 100 ng/mL). Interestingly, treatment with rh RANKL plus rh OPG resulted in an overall reduction in CCL20 mRNA expression compared with treatment with rh RANKL alone (Figure 4E). Notably, the lowest concentration of OPG (25 ng/mL) had the highest effect in significantly affecting chemokine mRNA expression whose levels returned to those found in untreated cultures (Figure 4E).

## 4. Discussion

In the present study, we found that increased serum levels of OPG at the end of gestation counteract the high circulating sRANKL levels present in patients with MS.

In addition, we reported for the first time that the placenta at term is an invaluable and prominent source of the homodimeric OPG, the most active form of the protein that is synthesized and released by the fetal organ.

Moreover, we showed that placental OPG interferes with the RANK–RANKL interaction, contrasting the up regulation of CCL20 mRNA expression in astrocytes.

The RANK and RANKL, a receptor–ligand pair of the TNFR superfamily, is a well-known system in bone remodeling/metabolism with emerging properties in other physiological processes including, osteoimmunology, organogenesis of the immune system, mammary glands development, and in the control of females’ body temperature in the central nervous system [32,33,34,35].

RANKL, either present in a soluble or membrane-bound form, binds the RANK receptor leading to the activation of several downstream signaling pathways that also mediate inflammatory response [36,37]. The interaction of RANK–RANKL is finely controlled by the decoy receptor OPG, that binding to RANKL makes it unavailable for interaction with its receptor, RANK [20].

OPG is also a TNFR family member, the only one to be produced as a soluble monomer or homodimer within the cells. The two forms of the protein are both released and active, however, the disulfide-linked dimer has a higher binding affinity for RANKL exerting more potent biological activity [38].

RANK and RANKL positive cells are commonly retrieved in a pro-inflammatory environment, and changes in the RANK-RANKL-OPG system have been associated with different autoimmune diseases in humans, including MS [21,39,40].

In particular, a significant decrease in OPG levels in cerebrospinal fluid or an increase of sRANKL in the plasma leading to a higher sRANKL/OPG ratio has been reported in MS patients at clinical onset and in advanced relapse remitting, respectively [41]. Similarly, we found higher levels of sRANKL in our MS females’ groups all of them in RRMS as compared to healthy female donors, and for the first time, we showed that pregnancy plays a key role in altering the ratio between circulating sRANKL and OPG in MS women.

We showed that the increase of serum OPG at term of gestation leads to an alteration of their ratio compatible with a decreased bioavailability of sRANKL, while, in the post-partum period, the OPG withdrawal that we report is consistent with the return of bioavailability of sRANKL to the non-pregnant condition. The relevance of this altered relationship that we have proven to occur during pregnancy is emphasized by the fact that the manipulation of circulating sRANKL/OPG ratio represents the current therapy for diseases in which the involvement of the RANK-RANKL system is well documented [42].

Furthermore, several endogenous factors are known to alter the sRANKL/OPG ratio favoring OPG expression [43], particularly noteworthy is vitamin D, whose reduction in the blood is considered a risk factor for developing MS and associated with MS disease activity [44].

Interestingly, the pregnancy-associated changes in sRANKL/OPG ratio that resulted from our study paralleled the evolution of the disease often observed in pregnant women with MS, who experience a reduction in relapse rate during late pregnancy and a resurgence in the post-partum period [1].

Pregnancy is definitely a time of major changes in maternal physiology that are largely orchestrated by the placenta, a dynamic organ that actively mediates the maternal adaptation to pregnancy by changing its structure and functionality with advancing in gestation [45].

The maternal MS does not associate with an altered uterine–placental interface or abnormalities in the placenta [46], suggesting that the placenta in pregnancies with multiple sclerosis is healthy and functional. Considering this, our results performed in the human placenta collected from healthy women also become relevant for MS pregnancies.

We provided evidence that the placenta expresses OPG with no difference in terms of protein expression level and distribution between the early gestation and term of pregnancy.

However, our data on the higher levels of OPG mRNA in term placenta as compared to the early gestation along with quantification of the protein in paired maternal and umbilical cord serum strongly indicate that the protein is highly produced and released at term of gestation by the placenta towards the maternal compartment.

According to this, our in vitro studies clearly demonstrated the pivotal role of the term placenta in the release of the OPG. Intriguingly, we have found that the fetal organ produces predominantly the homodimeric form of the protein known to increase of three orders of magnitude its affinity for RANKL, therefore, inhibiting the RANKL–RANK receptor interaction more powerfully [47].

In addition, the OPG dimer has greater ability than the monomeric one to distribute itself from the circulation to the peripheral compartments such as body tissues [48], suggesting that the OPG produced by the placenta could exert its biological activity far from the maternal–fetal interface.

Interestingly, Yano et al. have shown by using a homemade ELISA that OPG was present mainly as a monomer in human and murine blood [48,49]. They also reported the appearance of a homodimeric form of OPG in the serum of pregnant mice between day 8.5 and day 17.5 of pregnancy corresponding to the late second and third trimester of pregnancy in humans.

To understand factors able to regulate placental OPG production throughout gestation, we have performed in vitro experiments exposing placenta cultures to hormones characterizing the latest stages of pregnancy.

OPG is an estrogen-sensitive protein [50] and, by exposing the placenta explants to relevant concentrations of estradiol and estriol as it occurs in vivo at term of gestation, we found an increase in the release of the molecule in the conditioned medium with a more potent effect for the estriol versus the estradiol.

The hormonal milieu of the different stages of gestation is believed to be one of the potential mechanisms by which pregnancy exerts its beneficial effects on MS, based on this, important clinical trials have been developed by exogenously administered estrogens or progesterone with contrasting results [51]. However, it is necessary to underline that the administration of pregnancy-related sex hormones far from pregnancy did not account for the presence of the placenta. The placenta, in addition to producing estrogens, is also a target of these hormones, and it releases factors in the maternal blood according to the hormonal environment [52]. Thereby, placental factors including the placental OPG and not pregnancy-associated hormones by themselves could be responsible for the beneficial effects of pregnancy on MS.

Several systems and tissues of relevance for MS may benefit from OPG produced by the placenta; RANK-RANKL-OPG system participates in the regulation of the innate and adaptive immune response, and the elements of the triad are also expressed by microglia, astrocytes, neurons, and oligodendrocyte precursor cells [18,22].

Data from the literature suggest the RANK-RANKL-OPG system is part of the crosstalk and interactions between immune cells and CNS cells. In a mouse model of experimental autoimmune encephalomyelitis (EAE), the lack of RANKL in T cells or the astrocyte-specific deletion of RANK reduces the infiltration of T cells in the spinal cord and markedly protects mice from EAE [53].

According to the most accredited thesis while the passage across the blood–brain barrier (BBB) of auto-aggressive T-cells, activated in the periphery by CNS extrinsic factors is considered the triggering event, the interactions between activated immune cells and CNS cells would strengthen and perpetuate the inflammatory response within the CNS [54]. Pregnancy itself does not affect the integrity of the BBB [55], on the contrary, a breakdown of the barrier has been reported in MS patients [56]; so it is possible that during MS pregnancy, factors produced by a fully functional placenta such as the placental OPG may be active on tissues and cells of the pregnant mothers with effects either on immune cells of adaptive and innate immunity, cells within the CNS and/or on their reciprocal interaction.

Using astrocytes cells, we reported that placental OPG is able to significantly contrast the up-regulation of the chemokine CCL20 mRNA induced by recombinant RANKL that we have used to mimic the effect of a high or low bioavailability of sRANKL in accordance with our results in serum.

Chemokines are important molecules regulating leukocytes’ recruitment and infiltration into the tissues, and the CCL20 receptor, the CCR6 is selectively expressed by CD4+ T cells that produce the cytokine IL-17, i.e., Th17 [57]. CCL20 binding CCR6 induces the trafficking of Th17 cells into the CNS through the choroid plexus in the brain of EAE mice [58]. Additionally, cerebrospinal fluid and brain lesions are enriched by Th17 cells, and elevated CCL20 serum levels were found in MS patients [59,60].

Although from our experiments on conditioned media and OPG depletion, we cannot exclude that immunoprecipitation may have depleted the medium of other unknown factors, the data from rh OPG treatment support the role of placental OPG in reducing CCL20 mRNA expression. Interestingly our results obtained with recombinant OPG, supplied as the disulfide-linked homodimer, reporting a greater effect at the lowest doses are in line with Schneeweis et al., who clearly demonstrated that the high-affinity binding of OPG to RANKL relies on avidity, achieved only if OPG dimerizes [47]. Due to this, the production of a dimeric form by the placenta at the end of gestation is an event of great biological importance, through which the RANK-RANKL system could be physiologically and effectively inhibited in MS pregnancy.

Overall, based on our study, we propose a model (Figure 5) in which pregnancy thanks to the placenta promotes the production of a dimeric form of OPG that creates a conducive environment able to mitigate, acting on the sRANKL/OPG ratio, the excessive presence of sRANKL in peripheral blood of MS mothers with effects that can extend up to CNS on maternal astrocytes and their interaction with immune cells.

However, further research should be done to understand the precise functions of placental OPG, information that could reveal unique features of the protein produced by the fetal organ that may lead to developing better and/or new therapeutic strategies to treat MS patients.

## Figures and Tables

**Figure 1 cells-11-01357-f001:**
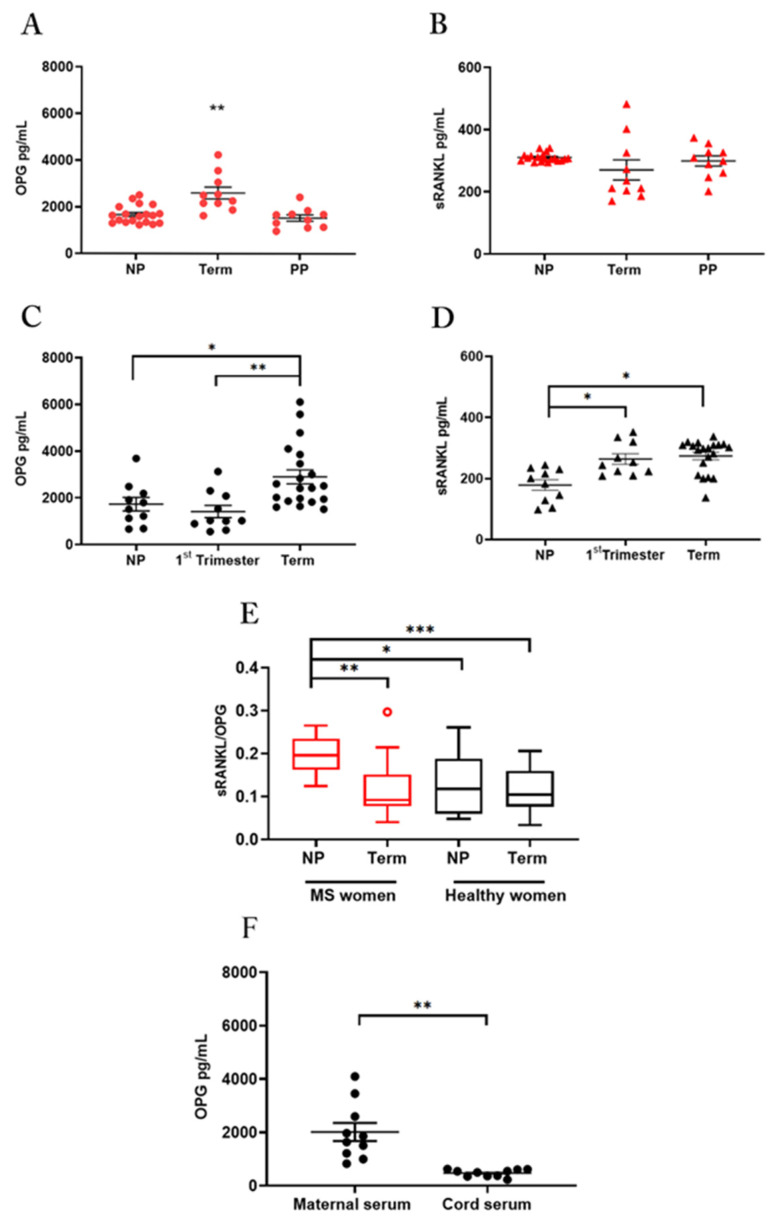
OPG and sRANKL in serum samples OPG and sRANKL in serum of MS women (**A**,**B**): ELISA assay for OPG (**A**) and sRANKL (**B**) in serum from non-pregnant women of reproductive age (NP; *n* = 19), at term of gestation (Term; *n* = 10) and in the post-partum period (PP; *n* = 10). OPG and sRANKL in serum of healthy women (**C**,**D**): ELISA assay for OPG (**C**) and sRANKL (**D**) in serum from healthy non-pregnant women of reproductive age (NP; *n* = 10), pregnant women, at first-trimester (1st Trimester; *n* = 10) and term of gestation (Term; *n* = 20). Serum sRANKL/OPG ratio in MS and healthy women (**E**): sRANKL/OPG ratio for MS patients, (non-pregnant: NP, *n* = 19; at term of gestation: Term, *n* = 10) and healthy women (non-pregnant: NP, *n* = 10; at term of gestation: Term, *n* = 20). Red circle in MS at term identifies one outlier. OPG quantification in paired maternal and cord serum (**F**): ELISA assay for OPG levels in paired maternal and cord serum (*n* = 10 donors). Each symbol represents a unique donor and mean ± SEM is shown for data (**A**–**D**) and (**F**). Data are presented as box-whisker plots in (**E**). Significance was determined with the Kruskal–Wallis test for multiple comparisons for data (**A**–**E**) and paired *t*-test for data in (**F**). * *p* < 0.05; ** *p* < 0.01; *** *p* < 0.001.

**Figure 2 cells-11-01357-f002:**
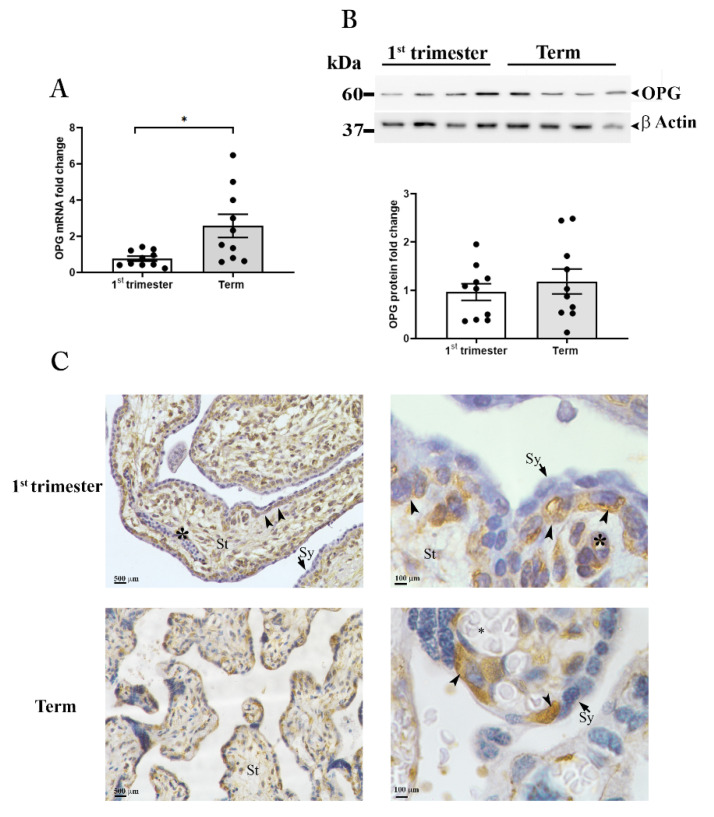
OPG expression in placental tissues. OPG mRNA expression (**A**) in placenta tissues at first-trimester (1st trimester; *n* = 10) and term (*n* = 10) pregnancy. (**B**) Representative Western blot and relative OPG protein expression in first-trimester (1st Trimester; *n* = 10) and term (Term; *n* = 10) placenta tissues. β-actin was used as loading control. (**C**) Representative OPG immunolocalization in first-trimester (upper panels) and term placenta tissues (lower panels) (*n* = 3 for each gestational period); villous stroma (St), perivascular cells of fetal capillaries (asterisk), cytotrophoblasts (arrowhead), syncytiotrophoblast (Sy, arrows). Data are presented as individual values, mean ± SEM are shown. Significance was determined with the unpaired *t*-test. * *p* < 0.05.

**Figure 3 cells-11-01357-f003:**
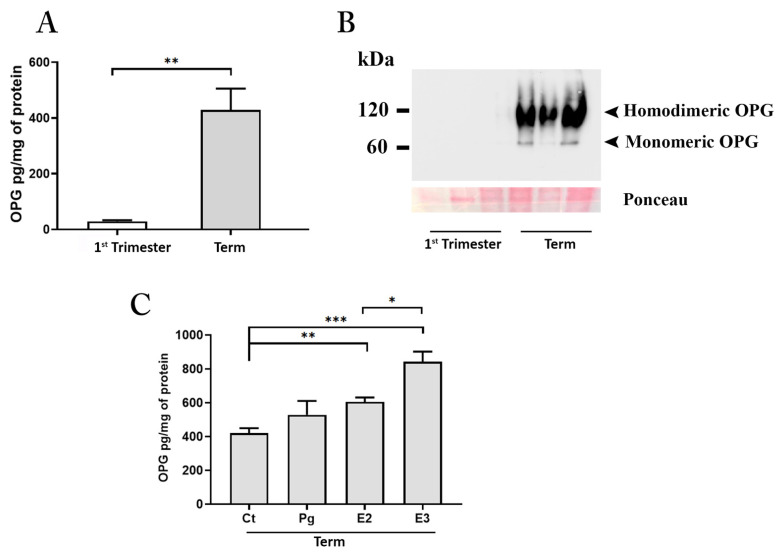
OPG production by placenta explants. ELISA assay for OPG in conditioned media (**A**) from first-trimester (maintained at 3% of oxygen for 48 h; *n* = 4 placentae at 9–11 weeks) and term (maintained at 8% of oxygen; *n* = 6 placentae at 38–39 weeks) placenta explants. Representative Western blot (**B**) under non-reducing conditions for OPG in conditioned media from first-trimester and term placenta explants. ELISA assay for OPG in conditioned media (**C**) from term placenta explants (maintained at 8% of oxygen; *n* = 3 placentae at 38–39 weeks) exposed for 48 h to estradiol (E2, 1 × 10^−8^ M), estriol (E3 7 × 10^−8^ M) and progesterone (Pg, 1 × 10^−6^ M) or medium alone (control, Ct). Data expressed as mean ± SEM. Significance was determined with the Unpaired *t*-test for OPG production in basal condition and one-way ANOVA for hormones exposure experiments. * *p* < 0.05; ** *p* < 0.01; *** *p* < 0.001.

**Figure 4 cells-11-01357-f004:**
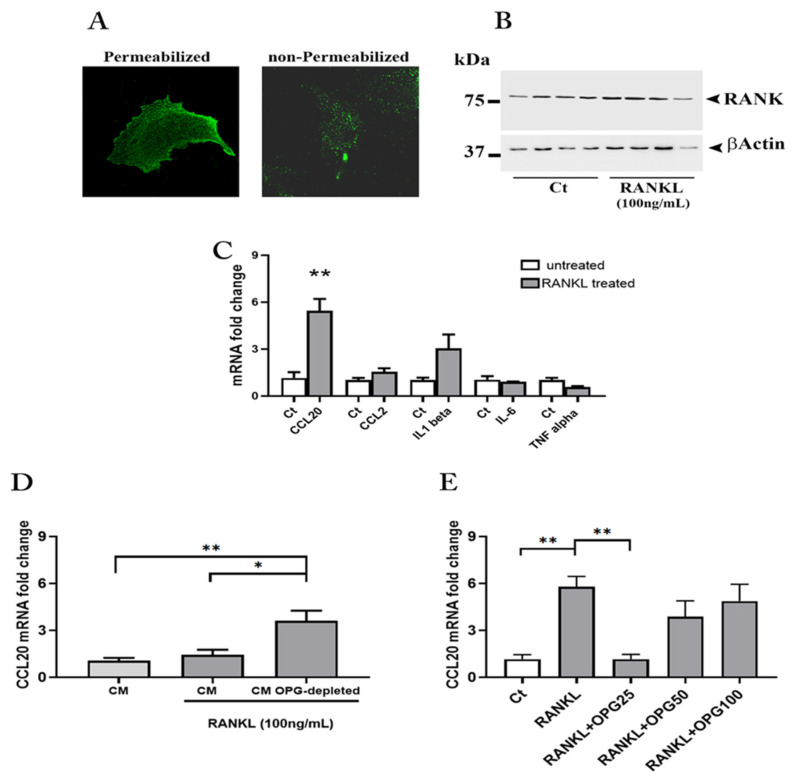
RANK-RANKL-OPG in human astrocytes. Representative immunofluorescence staining for RANK in astrocytes (**A**). RANK protein expression in astrocytes after the treatment with rh RANKL (original magnification 63X) (**B**). mRNA quantification for cytokines and chemokines in rh RANKL treated astrocytes (**C**). mRNA quantification for CCL20 in astrocytes treated with rh RANKL plus Conditioned Media (CM) from term placenta explants or CM OPG-depleted (**D**). mRNA quantification for CCL20 in astrocytes treated with rh RANKL or RANKL plus rh OPG (25, 50 and 100 ng/mL) (**E**). Data expressed as mean ± SEM of four separate experiments performed in quadruplicate. Significance was determined with Mann–Whitney test for data in (**C**) and one-way ANOVA for data in (**D**,**E**). * *p* < 0.05; ** *p* < 0.01.

**Figure 5 cells-11-01357-f005:**
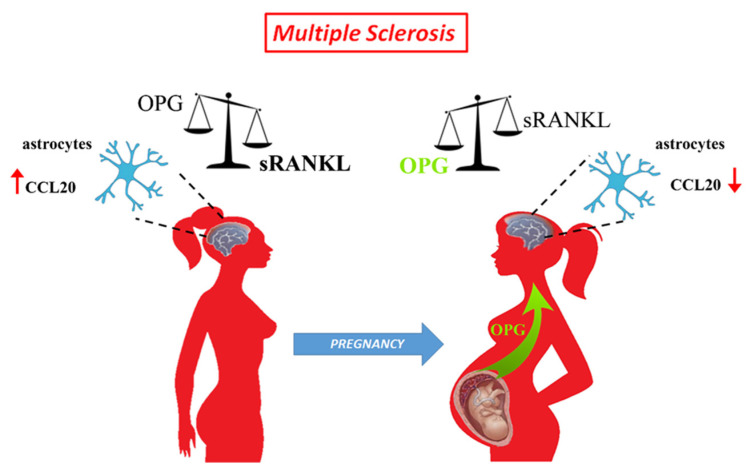
Schematic representation of the effect of pregnancy on sRANKL to OPG ratio in multiple sclerosis patients. MS patients have a dysregulated sRANKL to OPG ratio and increased expression of CCL20 within the CNS. During the third trimester of pregnancy, the placenta releases OPG in the maternal circulation. Placental OPG reverts the sRANKL/OPG balance, exerts inhibition on RANK-RANKL interaction within the maternal CNS, and leads to a reduction of CCL20 by astrocytes.

**Table 1 cells-11-01357-t001:** Healthy women.

Clinical Data	Not Pregnant (*n* = 10)	First-Trimester (*n* = 10)	Term (*n* = 20)
Maternal age	35.5 ± 2.0	28.2 ± 6.8	37.4 ± 9.0
Caucasian ethnicity (%)	-	100	100
Pre-pregnancy weight [mean (SD)]	-	-	64 ± 10.7
Weight gain (kg) [mean (SD)]	-	-	10.9 ± 8.8
Gestational age (weeks) [mean (SD)]	-	9 ± 2	38 ± 0.6
Nulliparous (%)	21	-	23.2
Female (%)	-	-	77
Male (%)	-	-	23
Placental weight (g) [mean (SD)]	-	12.3 ± 3.5	449.8 ± 83.6
Birth weight (g) [mean (SD)]	-	-	2879.1 ± 581

**Table 2 cells-11-01357-t002:** MS women.

Clinical Data	MS Women Pregnant (*n* = 10)	MS Women Non-Pregnant (*n* = 19)
Mean age (years) [mean (SD)]	34.9 ± 4.7	33.0 ± 8.4
Median number of relapses before pregnancy [median (range)]	2 (0–2.6)	-
Median number of relapses after delivery [median (range)]	0 (2–0)	-

## Data Availability

Not applicable.

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
