# Peer review of "Rank-Rankl-Opg Axis in Multiple Sclerosis: The Contribution of Placenta"

_cells, 2022, doi:10.3390/cells11081357_

Round 1

Reviewer 1 Report

The submitted research article analyses RANK-RANKL-OPG in the serum or placenta at different conditions, including Multiple Sclerosis patients. The manuscript is interesting, well organized and structured. The provided data supports the conclusion of the study, but several improvements are required for a better presentation.

I recommend to insert the number of the analysed samples in each experiment.

Insert references to previous works for E2, E3, PG, RANKL, OPG doses in in vitro cultures.

In each ELISA test include inter-intra-coefficient of variability, detection limits, replicate numbers and the number of the analysed samples.

The cohort described in par 2.2 for blood collection and processing does not correspond to the cohort described in figure 1 legend. Please, verify, add details and use the same abbreviation for the same groups.

FIGURE 2: insert B label for Western blot.

OPG Western blot in figure 2 display marked individual differences in term of OPG and housekeeping signals. How many samples were analysed? How do the authors explain these marked individual differences? What about the ratio between  OPG homodimeric and monomeric form?

OPG immunolocalization presented in figure 2c needs the inclusion control sections to asses the sprcificity of the immunoreaction (i.e. slides incubated without primary/secondary antiserum)

Figure 3a : What part of membrane was used for PounceauS staining ? Please set up the MW in figures. Pounceau S is not suitable for protein quantification , but it may be useful to indicate gel loading. In the provided figure, it is clear that 60kda OPG is not visible in gel lanes poorly stained with PonceauS.

Define abbreviatons at the first appearance in the main text

Author Response

Response to Referee 1.

The submitted research article analyses RANK-RANKL-OPG in the serum or placenta at different conditions, including Multiple Sclerosis patients. The manuscript is interesting, well organized and structured. The provided data supports the conclusion of the study, but several improvements are required for a better presentation.

I recommend to insert the number of the analyzed samples in each experiment.

  1. As suggested by the reviewer we inserted the number of samples analyzed for each experiment performed (see M&M)

Insert references to previous works for E2, E3, PG, RANKL, OPG doses in in vitro cultures.

  1. In the revised manuscript, we added the reference for RANKL and OPG treatment (Shang et al 2015; see M&M, section 2.4) and for hormones exposure experiments (Schock et al., 2016; see M&M section, 2.3)

In each ELISA test include inter-intra-coefficient of variability, detection limits, replicate numbers and the number of the analyzed samples. 

  1. We included in the revised manuscript the information required (see M&M, section 2.10).

The cohort described in par 2.2 for blood collection and processing does not correspond to the cohort described in figure 1 legend. Please, verify, add details and use the same abbreviation for the same groups.

  1. We have verified and corrected the mislabeled samples (See M&M, section 2.2 and legend of figure 1).

FIGURE 2: insert B label for Western blot.

  1. We have inserted the B label

OPG Western blot in figure 2 display marked individual differences in term of OPG and housekeeping signals. How many samples were analyzed? How do the authors explain these marked individual differences? What about the ratio between OPG homodimeric and monomeric form?

  1. The number of samples analyzed is n=10 for first trimester placentae and n=10 for term placentae, the number of samples analyzed is shown in the legend of Figure 2 and now also in M&M (see M&M, section 2.7). The differences in protein expression observed by the referee can be explained by inter-individual variability but also by the sex of the placenta. Our term placental tissues are for the most part females but we also have males, these clinical data are reported in table 1. For the first trimester placenta, we are not allowed for ethical issues, to perform genetic determination on the placental tissues thus the sex of samples is unknown. 

As concerns the placental expression of the homodimeric and monomeric forms of OPG, we have performed the native western blot in placental lysates, and the results obtained considering the ratio of the two bands (homodimeric and monomeric forms) for each sample showed a ratio of 1:1 for the first trimester and around 10:1 for the term placenta. However, the results obtained are essentially qualitative; indeed, the running condition and the % of acrylamide/bis acrylamide (8%) used to have a better resolution for the separation of proteins under non-reducing conditions did not allow us to use common housekeeping proteins, and the use of ponceau staining, as stated by the referee, is not a good procedure for quantification analysis. So, the only way to have consistent quantitative results to clarify the expression level of homodimeric and monomeric forms of OPG in the placenta is the use of specific antibodies for one of the two OPG forms that are not commercially available yet. 

OPG immunolocalization presented in figure 2c needs the inclusion control sections to assess the specificity of the immunoreaction (i.e. slides incubated without primary/secondary antiserum)

  1. We have included in supplementary material as Fig 1S the negative controls for first trimester and term placenta sections.

Figure 3a : What part of membrane was used for Ponceau staining ? Please set up the MW in figures. Ponceau S is not suitable for protein quantification , but it may be useful to indicate gel loading. In the provided figure, it is clear that 60 kDa OPG is not visible in gel lanes poorly stained with Ponceau.

  1. We agree with the referee that Ponceau staining is not suitable for protein quantification. We have reported ELISA on OPG content in placenta supernatants (Fig 3 A) that clearly showed the increased release of the protein in placenta cultures established from term placenta. The intent of the data presented in fig 3 B was to depict the qualitative difference in terms of OPG produced as a dimer by the placenta at the end of gestation vs the first trimester tissues. In the file uploaded see “original blots” we have shown the total membrane of the experiment, the cropped bands reported in fig 3B as well as different times of exposure. Following the referee's comment, we added the MW to the Ponceau band shown in Fig 3a.

Define abbreviations at the first appearance in the main text.

  1. We have revised it in accordance with the referee's comment.

Reviewer 2 Report

The authors have an exciting explanation for the reason why pregnant women with multiple sclerosis (MS) have a better condition during the term of pregnancy. They investigate the possible role of RANK-RANKL-OPG triad for the above phenomenon, pointing out that the placenta is providing the OPG secreted in an active, dimeric form. They measure the serum levels of OPG and sRANKL, carry out experiments with the placenta of different sources. In addition, they look at the effect of the above molecules on astrocytes, which have definite role in the progression of MS.

The paper is written in a well-understandable way, a planning of the experiment are well designed and easy to follow. The methods used are modern, the conclusion are acceptable.

My questions: Is it known what is the function of the placental OPG in healthy women? Is it the development/rearrangement of bones, the protection against inflammations, or both, or even more? It is claimed that the sRANKL/OPG ratio is the most important in the pregnant women’s blood for proper effect. I think in the in vitro experiments at the treatment of astrocytes with rhRANKL and OPG this ratio is much higher. Have the authors tried to use the “natural”, in vivo ratio of the components? Could it be possible to use cultures of placental cells from MS pregnant patients?

Author Response

Response to Referee 2.

The authors have an exciting explanation for the reason why pregnant women with multiple sclerosis (MS) have a better condition during the term of pregnancy. They investigate the possible role of RANK-RANKL-OPG triad for the above phenomenon, pointing out that the placenta is providing the OPG secreted in an active, dimeric form. They measure the serum levels of OPG and sRANKL, carry out experiments with the placenta of different sources. In addition, they look at the effect of the above molecules on astrocytes, which have definite role in the progression of MS.

The paper is written in a well-understandable way, a planning of the experiment are well designed and easy to follow. The methods used are modern, the conclusion are acceptable.

My questions: Is it known what is the function of the placental OPG in healthy women? Is it the development/rearrangement of bones, the protection against inflammations, or both, or even more?

  1. The high circulating level of OPG in the maternal serum has been associated with its action in the protection of maternal bone remodeling due to the high calcium request by the fetus, particularly in the third trimester of pregnancy. Studies on the role of placental OPG have recently shown a new function for the protein that would be involved in glucose homeostasis during pregnancy as reported by Huang et al 2020. A putative role for placental OPG in regulating inflammation at the maternal-fetal interface can be speculated on the basis of studies on RANKL/RANK axis in the decidua by Meng et al 2013 and 2017.

It is claimed that the sRANKL/OPG ratio is the most important in the pregnant women’s blood for proper effect. I think in the in vitro experiments at the treatment of astrocytes with rhRANKL and OPG this ratio is much higher. Have the authors tried to use the “natural”, in vivo ratio of the components?

  1. For the in vitro exposure experiments, we have selected the concentrations of sRANKL and OPG (both recombinant proteins) according to Shang et al 2015; we now added the reference to our manuscript (see M&M, section 2.4). The in vivo “natural” ratio is mimicked more closely by the treatment of astrocytes with placental conditioned media (CM), which, as shown in fig S3 (supplementary data), contains an amount of OPG ten-time greater than the concentration used for recombinant RANKL.

Could it be possible to use cultures of placental cells from MS pregnant patients?

  1. It is possible, although very difficult to obtain placentae from MS patients. Based on the literature, as we stated in the manuscript, no alteration in the placenta has been reported so far as well as placenta-related pregnancy disorders such as preeclampsia and IUGR are increased in MS pregnancy. All of this strongly suggests no relevant differences between the placenta of healthy women and those with MS and supports the use of placental samples from healthy pregnancies.